# Moesin Serves as Scaffold Protein for PD-L1 in Human Uterine Cervical Squamous Carcinoma Cells

**DOI:** 10.3390/jcm11133830

**Published:** 2022-07-01

**Authors:** Rina Doukuni, Takuro Kobori, Chihiro Tanaka, Mayuka Tameishi, Yoko Urashima, Takuya Ito, Tokio Obata

**Affiliations:** 1Laboratory of Clinical Pharmaceutics, Faculty of Pharmacy, Osaka Ohtani University, Tondabayashi 584-8540, Osaka, Japan; u4118098@osaka-ohtani.ac.jp (R.D.); u4117078@osaka-ohtani.ac.jp (C.T.); u4117083@osaka-ohtani.ac.jp (M.T.); urasiyo@osaka-ohtani.ac.jp (Y.U.); 2Laboratory of Natural Medicines, Faculty of Pharmacy, Osaka Ohtani University, Tondabayashi 584-8540, Osaka, Japan; itoutaku@osaka-ohtani.ac.jp

**Keywords:** programmed death ligand-1, ezrin/radixin/moesin, uterine cervical squamous cancer, immune checkpoint inhibitor, cancer immunotherapy

## Abstract

Immune checkpoint blockade (ICB) therapy targeting the programmed death ligand-1 (PD-L1)/PD-1 axis has emerged as a promising treatment for uterine cervical cancer; however, only a small subset of patients with uterine cervical squamous cell carcinoma (SCC) derives clinical benefit from ICB therapies. Thus, there is an urgent unmet medical need for novel therapeutic strategies to block the PD-L1/PD-1 axis in patients with uterine cervical SCC. Here, we investigated the involvement of ezrin/radixin/moesin (ERM) family scaffold proteins, which crosslink several plasma membrane proteins with the actin cytoskeleton, on the plasma membrane localization of PD-L1 in BOKU and HCS-2 cells derived from human uterine cervical SCC. Immunofluorescence analysis showed that PD-L1 colocalized with all three ERM proteins in the plasma membrane. Gene knockdown of moesin, but not ezrin and radixin, substantially reduced the plasma membrane expression of PD-L1, with limited effect on mRNA expression. An immunoprecipitation assay demonstrated the molecular interaction between PD-L1 and moesin. Moreover, phosphorylated, i.e., activated, moesin was highly colocalized with PD-L1 in the plasma membrane. In conclusion, moesin may be a scaffold protein responsible for the plasma membrane expression of PD-L1 in human uterine cervical SCC.

## 1. Introduction

Globally, uterine cervical cancer is the most common female genital tract cancer, ranked third in incidence and second in mortality, especially among female adolescents and young adults [1]. In the early stage, this cancer is treated effectively with surgical treatment and/or radiation therapy [2]. In contrast, the prognosis for patients diagnosed with recurrent and/or metastatic disease is particularly poor, with an extremely low 5-year survival rate due to limited responses to second-line treatments [3,4,5]. Moreover, there is no established standard second-line or later treatment options for patients with advanced stage.

Programmed death ligand-1 (PD-L1) is an immune checkpoint molecule expressed on a wide range of cancer cells that allows them to evade immunosurveillance via interaction with programmed death-1 (PD-1) [6]. Several clinical studies analyzing cervical cancer tissues have revealed that PD-L1 is highly expressed in cervical squamous cell carcinoma (SCC) [7,8,9,10]. Immune checkpoint blockade (ICB) therapy targeting the PD-1/PD-L1 axis enhances anti-tumor immunity by restoring anti-tumor T-cell-mediated immune responses [6]. Although ICB therapies have emerged as promising treatments for cervical cancer, only a small subset of patients with cervical SCC derives clinical benefit from ICB therapies [11,12,13]. Thus, there is an urgent unmet medical need for novel therapeutic strategies to block the PD-1/PD-L1 axis in patients with cervical SCC.

Accumulating evidence suggests that PD-L1 expression is regulated by several cellular processes, including transcriptional, post-transcriptional, and post-translational modifications [6,14,15,16]. As PD-L1 is a transmembrane protein, post-translational modifications, such as phosphorylation, glycosylation, and ubiquitination, play key roles in the cell surface membrane localization of PD-L1 [14,15,16].

Ezrin/radixin/moesin (ERM) protein family members are three closely related proteins distributed as scaffold proteins in a whole body [17]. Ezrin was discovered in chicken intestinal brush borders as a component of microvilli [18]. Radixin was purified from the undercoat of the adherens junction isolated from rat liver [19]. Moesin was isolated from bovine uteri enriched in smooth muscle cells as a heparin-binding protein [20]. ERM proteins consist of a 4.1-band ERM (FERM) domain at the amino-terminus, a central α-helical domain located between the amino-terminus and carboxy-terminus domains, and an F-actin-binding domain at the carboxy-terminus [21]. The FERM domain normally interacts with the intramolecular carboxy-terminus in the cytosol, leading to dormant inactive conformation of ERM proteins that is incapable of interacting with other proteins; while the intramolecular binding is dissociated by phosphorylation of threonine on the carboxy-terminus [21,22,23], enabling ERM proteins to have active conformation that can play multiple roles in cell motility, cell adhesion, and signal transduction [17,24,25,26]. Furthermore, ERM proteins serve as crosslinkers between several cancer-related transmembrane proteins, including several transmembrane receptor kinases, and the actin cytoskeleton network, contributing to their cell surface localization [27,28,29]. We have recently reported that ERM proteins post-translationally regulate the cell surface localization of PD-L1 by serving as scaffold proteins in different manners in several types of human cancer cells [30,31,32,33,34,35]. However, it remained unclear whether ERM proteins are involved in the cell surface plasma membrane localization of PD-L1 in cervical SCC.

In this study, we investigated the involvement of ERM proteins in the plasma membrane localization of PD-L1 using BOKU cells derived from human cervical SCC.

## 2. Materials and Methods

### 2.1. Cell Culture

The human cervical SCC cell lines, BOKU (IFO50323) established by Nozawa et al. and HCS-2 (JCRB1203) established by Morisawa et al. [36] were both purchased from the Japanese Collection of Research Bioresources Cell Bank (Osaka, Japan). The BOKU cells were grown in Dulbecco’s modified Eagle’s medium (DMEM) containing L-glutamine and phenol red (041-29775; FUJIFILM Wako Pure Chemical, Osaka, Japan) and HCS-2 cells were cultured in Ham’s F-12 medium containing L-glutamine and phenol red (087-08335; FUJIFILM Wako Pure Chemical), both supplemented with heat-inactivated 10% fetal bovine serum (BioWest, Nuaille, France) in culture flasks until 70–80% confluence and were maintained at 37 °C in a humidified atmosphere with 5% CO_2_.

### 2.2. Small Interfering (si)RNA Treatment

BOKU cells and HCS-2 cells were seeded in 24-well cell culture plates at a density of 2.0 × 10^4^ cells/well for total RNA extraction and flow cytometry, or in 96-well cell culture plates at a density of 4.0 × 10^3^ cells/well for cell viability assays, respectively, and incubated overnight to allow cell attachment. The siRNAs targeting human genes of interest or nontargeting control (NC) were introduced into cells at a concentration of 5 nM using Lipofectamine RNAiMAX at doses of 0.1 µL/1.0 × 10^4^ cells. After the addition of siRNA and the transfection reagent, the cells were cultured continuously for four days (BOKU cells) or three days (HCS-2 cells) without medium replacement. All reagents used for siRNA treatment were purchased from Thermo Fisher Scientific (Tokyo, Japan).

### 2.3. Real-Time Reverse Transcription Polymerase Chain Reaction (RT-PCR)

Total RNA was extracted from cells using an ISOSPIN Cell and Tissue RNA kit (Nippon Gene, Tokyo, Japan) according to the manufacturer’s protocol. The RNA concentration and purity were evaluated using a NanoDrop Lite spectrophotometer (Thermo Fisher Scientific). RT-PCR and relative transcript quantification were performed as described previously [30,32]. The sequences of the gene-specific PCR primers used are listed in Appendix A. All reagents and equipment used for RT-PCR were obtained from TaKaRa Bio (Shiga, Japan) and Bio-Rad Laboratories (Hercules, CA, USA), respectively.

### 2.4. Confocal Laser-Scanning Microscopy (CLSM)

Cells plated on a polylysine-coated 35-mm glass-bottom dish (Matsunami Glass, Osaka, Japan) at a density of 1.0 × 10^5^ cells were incubated at 37 °C overnight in a humidified atmosphere with 5% CO_2_. The cells were fixed with 4% paraformaldehyde, permeabilized with 0.5% Triton-X100, and blocked with Dulbecco’s phosphate-buffered saline (D-PBS) supplemented with 0.3 M glycine, 10% normal goat serum, 1% bovine serum albumin, and 0.1% Tween-20. Next, the cells were incubated with primary antibodies (Abs) against PD-L1, ezrin, radixin, and moesin at 4 °C overnight and then labeled with an Alexa Fluor 488-conjugated secondary Ab for ezrin, radixin, and moesin at room temperature for 1 h. Subsequently, the cellular membrane was counterstained with tetramethylrhodamine isothiocyanate-conjugated phalloidin, a high-affinity F-actin probe. The preserved cells were observed and photographed at 0.5–1.0 µm intervals on the *z*-axis at an original magnification of 20× using a Nikon Al CLSM system and NIS-Elements AR software (Nikon Instrument, Tokyo, Japan).

For double immunofluorescence staining, the procedure before the secondary Ab reaction was the same as described above. Thereafter, the cells were labeled with an Alexa Fluor 594-conjugated secondary Ab at room temperature for 1 h, and then with an Alexa Fluor 488-conjugated anti-PD-L1 Ab at 4 °C overnight. The subsequent procedure was conducted as described above. All Abs used in this study are listed in Appendix A. No fluorescence signals were observed in cells incubated with goat anti-rabbit IgG secondary antibodies conjugated with Alexa Fluor 488 or Alexa Fluor 594 without primary antibodies (Appendix A).

### 2.5. Cell Viability Assay

The cells were treated with siRNAs targeting the genes of interest or 1.0 µM of staurosporine (Merck, Darmstadt, Germany), a suitable positive control to significantly decrease in vitro cell viability, for four days, without medium exchange. Subsequently, the cells were incubated with PrestoBlue Cell Viability Reagent (Thermo Fisher Scientific), a fast and sensitive assay for assessing cell viability [37,38] at 37 °C for 10 min under humidified conditions with 5% CO_2_, protected from direct light. Then, fluorescence signals were detected at wavelengths of 560 nm (excitation) and 590 nm (emission) using a Synergy HTX Multi-Mode Microplate Reader (BioTek Instrument, Winooski, VT, USA).

### 2.6. Western Blotting

Cells were lysed in radioimmunoprecipitation assay (RIPA) buffer containing a protease inhibitor cocktail on ice. Lysates containing equal amounts of proteins were heated at 97 °C for 5 min in 2× sample buffer consisting of 0.125 M Tris-HCl, 4% sodium dodecyl sulfate (SDS), 20% glycerin, 0.01% bromophenol blue, and 10% 2-mercaptoethanol. The proteins were separated by SDS-polyacrylamide gel electrophoresis and transferred to nitrocellulose membranes. The membranes were blocked with 5% skim milk in PBS with 0.1% Tween-20, and then probed with the respective primary Abs at 4 °C overnight. The membranes were incubated with the respective horseradish peroxidase-conjugated secondary Abs, and visualized with an enhanced chemiluminescence system on a LuminoGraphII EM (ATTO, Tokyo, Japan). All original immunoblot images are provided in Appendix A.

### 2.7. Immunoprecipitation

Immunoprecipitation assays were conducted as previously described [30,31,32,33], with some modifications. Briefly, 500 μL of whole-cell lysates collected in RIPA buffer containing a protease inhibitor cocktail was incubated with 50 μL of protein A beads (nProtein A Sepharose 4 Fast Flow; Cytiva, Tokyo, Japan) on a rotating wheel at 4 °C for 1 h to eliminate non-specific interactions. The pre-cleaned lysates were incubated with an anti-PD-L1 Ab or control IgG Ab on a rotating wheel at 4 °C overnight, and then mixed with 50 μL of protein A beads on a rotating wheel at 4 °C for 3 h. Next, the protein A beads were collected and rinsed three times with RIPA buffer, followed by heating at 97 °C for 5 min in 2× sample buffer (NacalaiTesque, Kyoto, Japan) before immunoblotting.

### 2.8. Flow Cytometry

Flow cytometry analysis was performed as previously described [30,31,32,33], with some modifications. Single-cell suspensions were reacted with an allophycocyanin (APC)-conjugated anti-PD-L1 Ab (2.8 μg/tube) in a labeling buffer consisting of D-PBS, 5% normal horse serum, and 1% sodium azide at 4 °C for 1 h. Subsequently, the mean fluorescence intensity of APC-PD-L1 on the cell surface was analyzed using an EC800 Flow Cytometry Analyzer and EC800 software (Sony Imaging Products and Solutions, Tokyo, Japan).

### 2.9. Analysis for Gene Expression of PD-L1 and the ERM Family in Patients with Cervical SCC

Gene expression levels PD-L1 and the ERM family in the tumor tissues derived from patients with cervical SCC were assessed with UALCAN, as were the impacts of moesin expression levels on the patient survival probabilities. UALCAN is a web portal that enables the evaluation of gene expression and survival analysis on approximately 20,500 protein-coding genes in 33 different tumor types using RNA-sequencing data obtained from The Cancer Genome Atlas (TCGA) project [39,40].

### 2.10. Statistical Analysis

Data are shown as the mean ± standard error of the mean (SEM). Statistical analysis was performed using Prism software version 3 (GraphPad Software, La Jolla, CA, USA). Statistical significance was evaluated by one-way analysis of variance (ANOVA) followed by Dunnett’s test for multiple comparisons. Differences were considered significant at *p* < 0.05.

## 3. Results

### 3.1. Gene and Protein Expression Profiles of PD-L1 and the ERM Family in the Human Uterine Cervical Squamous Cancer Cells

We analyzed the gene expression profiles of PD-L1 and the ERM family members in several human uterine cervical squamous cancer cell lines registered in the public database of the Cancer Cell Line Encyclopedia (CCLE) and the Cancer Dependency Map (DepMap) portal data explorer. For BOKU cells, the database analysis revealed higher mRNA levels of PD-L1 and ezrin, and moderate mRNA levels of radixin and moesin, respectively, compared to other cell types (Figure 1a). Although the relative mRNA expression level of PD-L1 was low in HCS-2 cell, those of ezrin, radixin, and moesin in HCS-2 cells rank high among human uterine cervical squamous cancer cell lines analyzed (Figure 1a).

Next, the relative mRNA and protein expression levels of PD-L1, ezrin, radixin, and moesin in BOKU cells and HCS-2 cells were examined using RT-PCR and western blotting, respectively. PD-L1, ezrin, radixin, and moesin mRNA and protein expression were detected in BOKU cells (Figure 1b,c) and also in HCS-2 cells (Appendix A).

### 3.2. Plasma Membrane Localization of PD-L1 and the ERM Family in the Human Uterine Cervical Squamous Cancer Cells

The subcellular localization of PD-L1 and ERM proteins in BOKU cells and HCS-2 cells was scrutinized using immunofluorescence CLSM. The fluorescence signals of PD-L1, ezrin, radixin, and moesin in BOKU cells (Figure 2a–d) and HCS-2 cells (Appendix A) were highly colocalized with that of F-actin, a representative cellular plasma membrane marker, implicating specific plasma membrane localization of these proteins in the human uterine cervical squamous cancer cells. Interestingly, double immunofluorescence staining results demonstrated that PD-L1 was highly colocalized with the ERM proteins, especially in the plasma membrane of BOKU cells (Figure 3a–c) and HCS-2 cells (Appendix A).

### 3.3. Effects of siRNA-Mediated Knockdown of the ERM Family on Target mRNA Levels and Cell Viability in the Human Uterine Cervical Squamous Cancer Cells

Next, we checked the effects of siRNAs against the ERM members on the expression of each target gene and on the viability of the human uterine cervical squamous cancer cells. Treatment of BOKU cells and HCS-2 cells with siRNAs against ezrin, radixin, and moesin effectively significantly suppressed the mRNA levels of the respective targets (Figure 4a–c and Appendix A), without affecting the cell viability of BOKU cells (Figure 4d) and HCS-2 cells (Appendix A).

### 3.4. Effects of siRNAs against ERM on the mRNA and Cell Surface Expression Levels of PD-L1 and the Molecular Interaction of PD-L1 and Moesin in the Human Uterine Cervical Squamous Cancer Cells

We evaluated the effects of knockdown of ezrin, radixin, and moesin on the mRNA and cell surface expression levels of PD-L1 in BOKU cells and HCS-2 cells. Treatments of cells with the ERM siRNAs had no effects on the mRNA levels of PD-L1 as compared with NC siRNA (Figure 5a and Appendix A). Flow-cytometric analysis showed that moesin siRNA, but not ezrin and radixin siRNAs, significantly reduced the cell surface expression of PD-L1 to the same extent as PD-L1 siRNA did in BOKU cells (Figure 5b,c) and HCS-2 cells (Appendix A). Co-immunoprecipitation assays demonstrated the molecular interaction between PD-L1 and moesin in BOKU cell (Figure 5d) and HCS-2 cells (Appendix A) in addition to the molecular interaction between PD-L1 and ezrin, radixin in BOKU cells (Appendix A). These findings indicated that moesin may contribute to the plasma membrane localization of PD-L1 in the human uterine cervical squamous cancer cells, probably via post-translational modifications.

### 3.5. PD-L1 Colocalizes with Phosphorylated Moesin in BOKU Cells

Next, we analyzed whether phosphorylated, i.e., activated, moesin (p-moesin) colocalizes with PD-L1 in BOKU cells using immunofluorescence staining. Intriguingly, we found that p-moesin was highly colocalized with PD-L1 in the plasma membrane of BOKU cells (Figure 6).

### 3.6. Gene Expression Analysis of PD-L1 and the ERM Family in Human Cervical SCC Tissue

Finally, we assessed gene expression levels PD-L1 and the ERM family in tumor tissues derived from patients with cervical SCC. Gene expression of PD-L1 was significantly higher in the human cervical SCC tissues relative to normal tissues, while there were no significant changes in the gene expression of the ERM family (Figure 7a). In contrast, no correlations were observed between PD-L1 and three ERM gene expressions. Importantly, survival probability in the cervical SCC patients with higher expression of moesin was not significantly lower (*p* = 0.083) albeit clearly lower than those with low/medium expression of moesin (Figure 7b).

## 4. Discussion

This study demonstrates for the first time that BOKU human cervical SCC cells abundantly express not only PD-L1, but also all three ERM family members at the mRNA and protein levels; and that PD-L1 is highly colocalized with all three ERM members in the plasma membrane due to their specific localization in the plasma membrane. Additionally, we also verified the similar results in HCS-2 cells. Many studies in human clinical SCC tissues have revealed that PD-L1, ezrin, and moesin are widely expressed in laryngeal [41,42,43] and oral [44,45,46,47,48,49] SCCs, PD-L1; and that ezrin are expressed in tongue [50,51,52] and esophageal [53,54,55,56,57] SCCs; and that PD-L1 and moesin are expressed in uterine cervix SCC [58,59,60]. In contrast, radixin expression has been reported only in the human tongue SCC cell line Tca8113 [61]. Together, the present results demonstrate that in BOKU cells, PD-L1 and all three ERM proteins are expressed at relevant levels and are predominantly colocalized in the cellular plasma membrane, in line with the previous observations in human clinical SCC tissues.

As post-translational factors, ERM proteins contribute to the cell surface localization of several cancer-related transmembrane proteins, such as epidermal growth factor receptor 2 and several drug transporters, by cross-linking them with the actin cytoskeleton [27,29,62,63]. Interestingly, our recent studies demonstrated that among the ERM proteins, ezrin predominantly controls the plasma membrane localization of PD-L1, possibly via molecular interaction, in human adenocarcinoma cells derived from uterine cervix (HeLa) and colorectum (LS180) cells, in which ezrin is expressed at the highest level among ERM, based on DepMap global gene expression analysis and RT-PCR data [31,32]. We also found that as a dominant ERM protein, radixin acts as a principal scaffold protein for PD-L1 in KP-2 human pancreatic ductal adenocarcinoma cells [30]. Significantly, siRNA-mediated interference of moesin, but not ezrin and radixin, greatly reduced the cell surface expression of PD-L1, with limited effect on the PD-L1 mRNA level. These results imply that moesin may be a dominant ERM protein responsible for the cell surface localization of PD-L1 in BOKU cells and HCS-2 cells, in which all three ERM proteins are highly expressed. This idea may be supported by the findings that PD-L1 was co-immunoprecipitated with moesin and that PD-L1 was highly colocalized with p-moesin, i.e., the activated form of moesin, in BOKU cells. Recently, Meng et al. reported that in the human breast cancer cell line, p-moesin prevents PD-L1 from the ubiquitination that leads to the proteasomal degradation by competing with E3 ubiquitin ligase, thereby inhibiting degradation of PD-L1, which in turn stabilizes PD-L1 in the surface plasma membrane [64]. These observations raise the possibility that p-moesin may contribute to the plasma membrane stabilization of PD-L1 in BOKU cells, possibly by preventing the ubiquitin-mediated proteasomal degradation. On the other hand, the mechanism by which ERM proteins differently influence the plasma membrane localization of PD-L1 in several cancer cell types remains unknown. One possibility is that among ERM proteins, the principal partner for PD-L1 may be attributed, at least in part, to a large variation of ERM expression profile according to the histological cancer types and/or cancer sites. A growing body of evidence suggests that radixin predominantly regulates the plasma membrane localization of multidrug resistance protein 2, an efflux transporter abundant in hepatocyte [65,66,67,68,69,70]. This is likely, because radixin is a dominant ERM protein in the hepatic tissues and hepatocytes [71]. Given that in contrast to ezrin and radixin, moesin is highly expressed in the human uterine cervix SCC [58,59,60], moesin may regulate the plasma membrane localization of PD-L1 as a predominant ERM protein in BOKU cells and HCS-2 cells, as is the case with ezrin in HeLa and LS180 cells [31,32]. The detailed mechanism by which PD-L1 selects a particular ERM protein as a partner for its plasma membrane localization in several cancer cell types will have to be addressed in future studies.

Importantly, the expression statuses of moesin and PD-L1 have been associated with poor prognosis, tumor malignancy, and metastasis in patients with uterine cervical SCC [58,59,60,72]. In fact, analysis using the human cervical SCC tissues obtained from TCGA database showed a higher expression of PD-L1 gene in the human cervical SCC tissue and a lower survival rate in the cervical SCC patients who had a higher expression of moesin. These clinical data may partly support our hypothesis that there is a possible association between PD-L1 and moesin in uterine cervical SCC cells.

Our therapeutic strategy that siRNA-mediated inhibition of moesin decreases the cell surface localization of PD-L1 from intracellular compartments seems to be an attractive therapeutic modality independent of the mutation in the extracellular region of PD-L1 protein, where the target sequence of Abs against PD-L1 is present and is the cause of tolerance to the PD-L1 Abs [73]. Therefore, one possible advantage of this method is that it can be applicable to patients who develop tolerance to the existing Abs against PD-L1.

In order to ascertain whether the results of our present in vitro data are also observed in in vivo experiments employing xenograft model mice should be performed in our future studies.

## 5. Conclusions

In summary, using BOKU cells, we demonstrated for the first time that PD-L1 and all three ERM proteins are abundantly expressed in uterine cervical SCC cells and highly colocalized in the plasma membrane. Among the three ERM proteins, moesin may act as an essential scaffold protein predominantly responsible for the plasma membrane localization of PD-L1 through the protein–protein interaction in BOKU cells via post-translational modification.

The limitation of this study is that in vitro relationship between PD-L1 and moesin in BOKU cells does not fully mimic clinical patients with uterine cervical SCC who have received ICB therapy. Additionally, we only used two human uterine cervical SCC cell lines, although there exist many more kinds of human uterine cervical SCC cell lines. Therefore, in future research, we should address these issues with other human uterine cervical SCC cell lines and also with in vivo xenograft model mice to demonstrate the relationship between PD-L1 and moesin in human uterine cervical SCC.

## Figures and Tables

**Figure 1 jcm-11-03830-f001:**
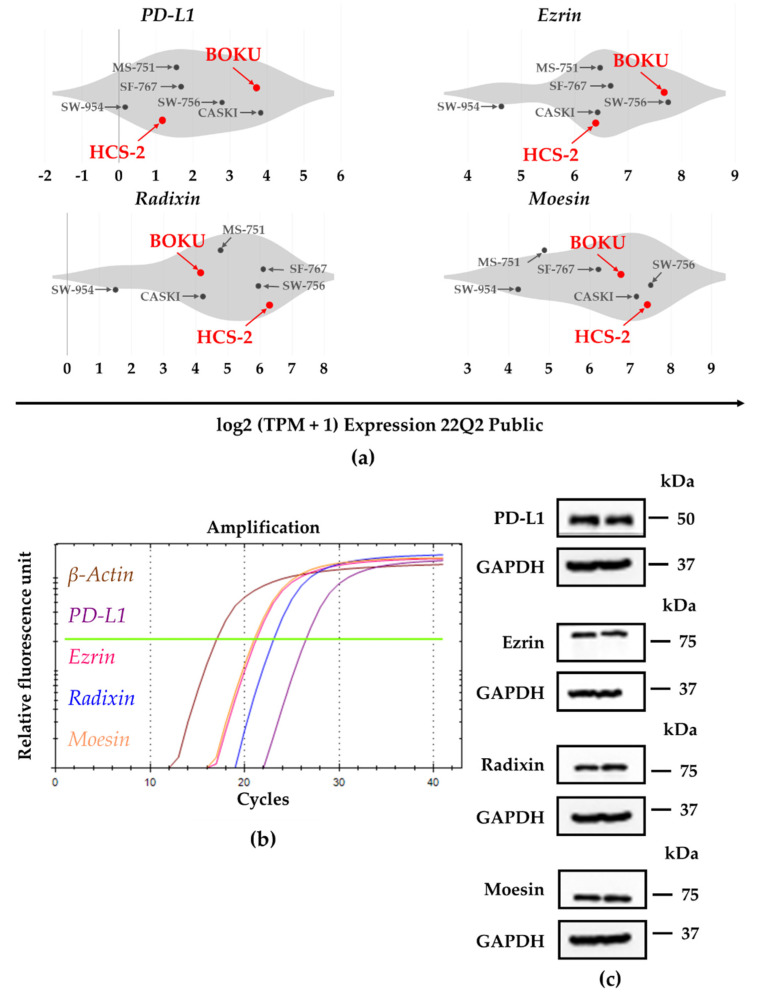
Gene and protein expression profiles of programmed death ligand-1 (PD-L1), ezrin, radixin, and moesin (ERM) in the human uterine cervical squamous cancer cells. (**a**) Violin plots showing the median gene expression (log2 (TPM +1)) of PD-L1 and the ERM family members in various human uterine cervical squamous cancer cell lines evaluated by utilizing the Cancer Dependency Map (DepMap), Broad (2022): DepMap 22Q1 Public. (**b**) Representative amplification curves for PD-L1, ezrin, radixin, and moesin as well as β-actin, indicating relative mRNA expression in BOKU cells as determined by real-time reverse transcription polymerase chain reaction (RT-PCR). (**c**) Representative western blot images for PD-L1, ezrin, radixin, moesin, and glyceraldehyde-3-phosphate dehydrogenase (GAPDH), indicating protein levels in whole-cell lysates of BOKU cells. Molecular weights are denoted in kDa. The data are representative of three independent experiments using at least three independent samples of total RNA and protein extracts.

**Figure 2 jcm-11-03830-f002:**
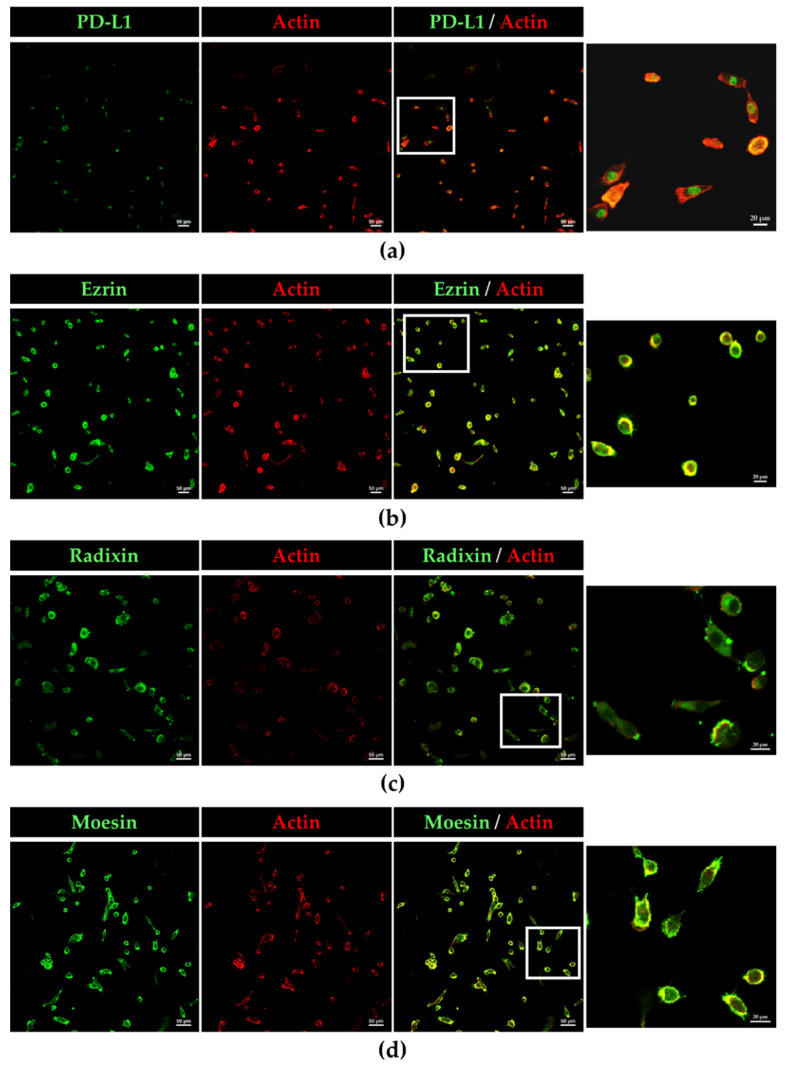
Confocal laser scanning microscopy (CLSM) analysis of the subcellular distribution of PD-L1, ezrin, radixin, and moesin in BOKU cells. (**a**) PD-L 1, (**b**) ezrin, (**c**) radixin, and (**d**) moesin labeled with Alexa Fluor 488 (green) highly colocalized with F-actin labeled with tetramethylrhodamine (red). Scale bars: 50 µm. Higher magnification images in the rightmost are from the corresponding white rectangle region in the merged panels. Scale bars: 20 µm. All images are representative of at least three independent experiments.

**Figure 3 jcm-11-03830-f003:**
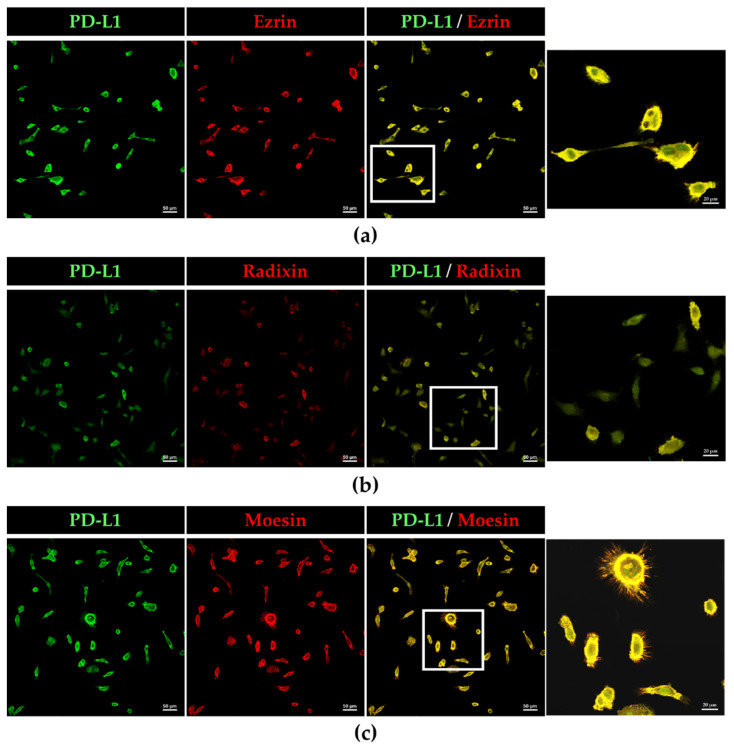
Colocalization of PD-L1 with ezrin, radixin, and moesin in the plasma membrane of BOKU cells. CLSM revealed that PD-L1 labeled with Alexa Fluor 488 (green) was predominantly colocalized with (**a**) ezrin, (**b**) radixin, and (**c**) moesin, labeled with Alexa Fluor 594 (red) in the plasma membrane. Scale bars: 50 µm. Higher magnification images in the rightmost are from the corresponding white rectangle region in the merged panels. Scale bars: 20 µm. All images are representative of at least three independent experiments.

**Figure 4 jcm-11-03830-f004:**
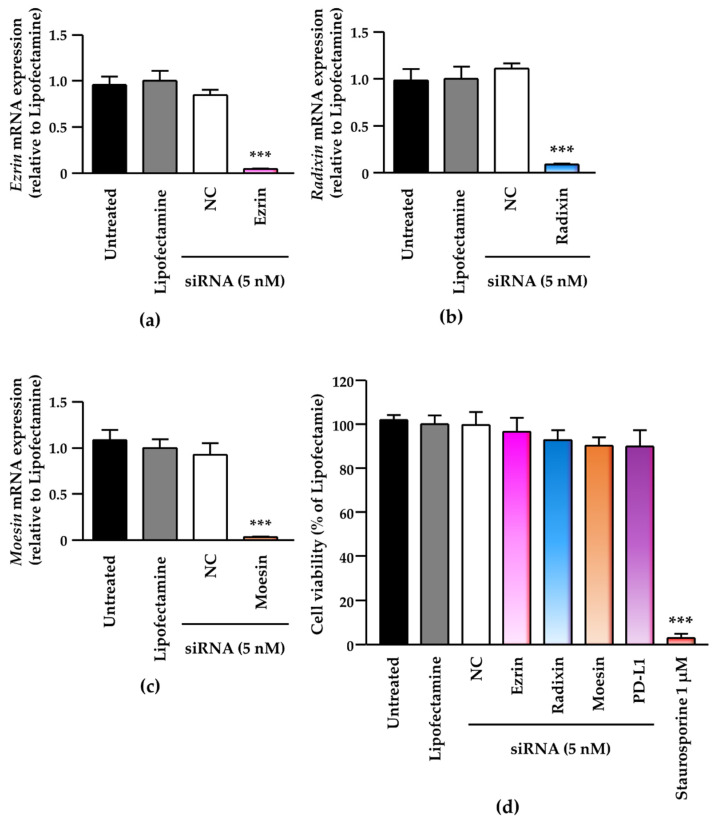
Effects of small interfering (si) RNA-mediated knockdown of the ERM family on target mRNA levels and cell viability in BOKU cells. Cells were treated with transfection medium (Untreated), transfection reagent (Lipofectamine), nontargeting control (NC) siRNA, or specific siRNAs for ezrin, radixin, or moesin at 5 nM, and then cultured for four days. Each column shows the mRNA level of (**a**) ezrin, (**b**) radixin, and (**c**) moesin normalized to β-actin in cells from each treatment group relative to that in cells treated with Lipofectamine alone, as determined by RT-PCR. *n* = 3, *** *p* < 0.001 vs. Lipofectamine. All data are expressed as the mean ± standard of the mean (SEM) and were analyzed by one-way analysis of variance (ANOVA) followed by Dunnett’s test. (**d**) Cell viability of BOKU cells. *n* = 6, *** *p* < 0.001 vs. Lipofectamine. All data are expressed as the mean ± SEM and were analyzed by one-way ANOVA followed by Dunnett’s test.

**Figure 5 jcm-11-03830-f005:**
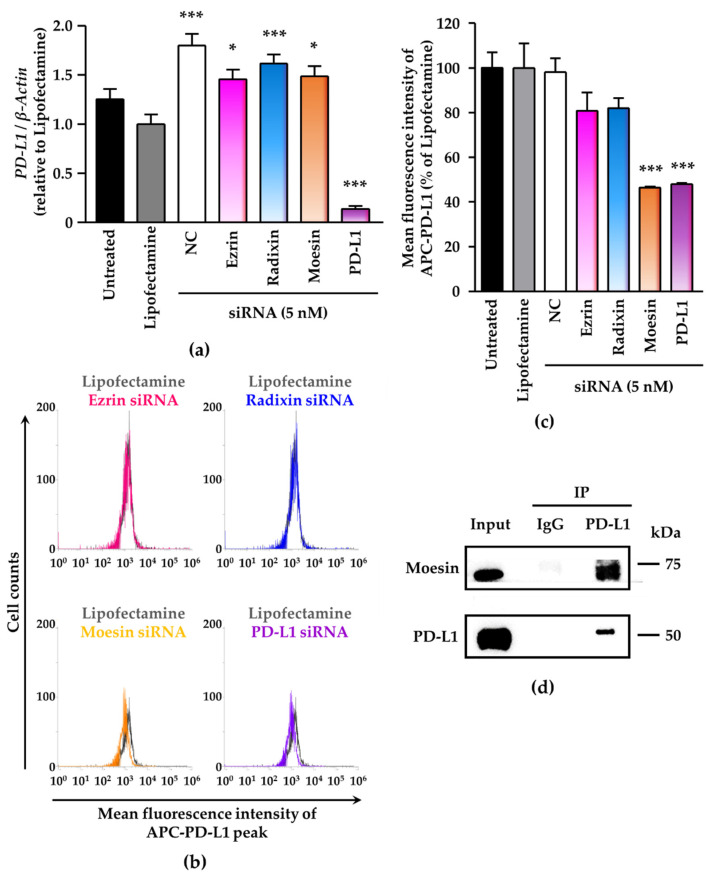
Contribution of moesin to PD-L1 cell surface expression in BOKU cells. Cells were cultured with transfection medium (Untreated), transfection reagent (Lipofectamine), NC siRNA, or the ERM siRNAs for four days (**a**–**c**). (**a**) PD-L1 mRNA levels normalized to β-actin in cells in all treatment groups relative to that in Lipofectamine-treated cells were determined by RT-PCR; *n* = 3, *** *p* < 0.001, * *p* < 0.05 vs. Lipofectamine. All data are expressed as mean ± SEM and were analyzed by one-way ANOVA followed by Dunnett’s test. (**b**) Overlay of representative histograms of the mean fluorescence intensity of allophycocyanin (APC)-labeled PD-L1 on the surface plasma membrane of BOKU cells as measured by flow cytometry. (**c**) Mean fluorescence intensity of APC-PD-L1 on the surface plasma membrane, quantified by flow cytometry, for all treatments relative to Lipofectamine treatment; *n* = 3, *** *p* < 0.001 vs. Lipofectamine. All data are expressed as mean ± SEM and were analyzed by one-way ANOVA followed by Dunnett’s test. (**d**) Representative immunoblots of moesin and PD-L1 in whole-cell lysates (input) and immune precipitates pulled down by a control IgG or an anti-PD-L1 antibody. Molecular weight is expressed in kDa.

**Figure 6 jcm-11-03830-f006:**
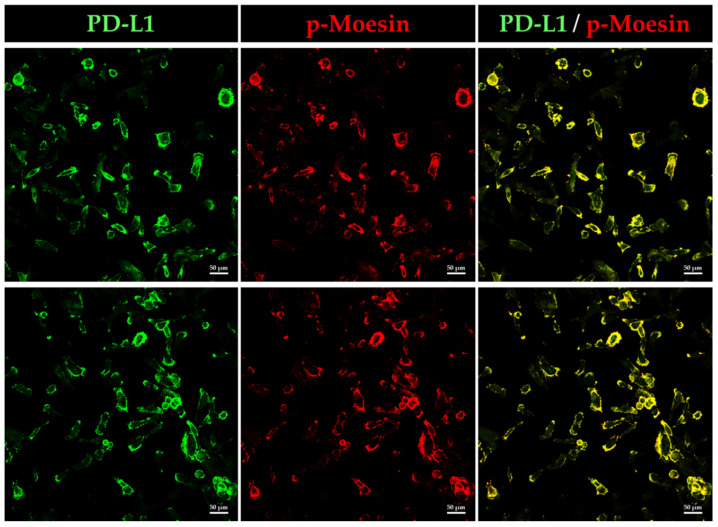
Phosphorylated moesin (p-moesin) colocalizes with PD-L1 in BOKU cells. Subcellular localization of p-moesin and PD-L1 in BOKU cells as observed by CLSM. PD-L1 labeled with Alexa Fluor 488 (green) was highly colocalized on the plasma membrane with p-moesin labeled with Alexa Fluor 594 (red). Scale bars: 50 μm. Data are representative of at least three independent experiments.

**Figure 7 jcm-11-03830-f007:**
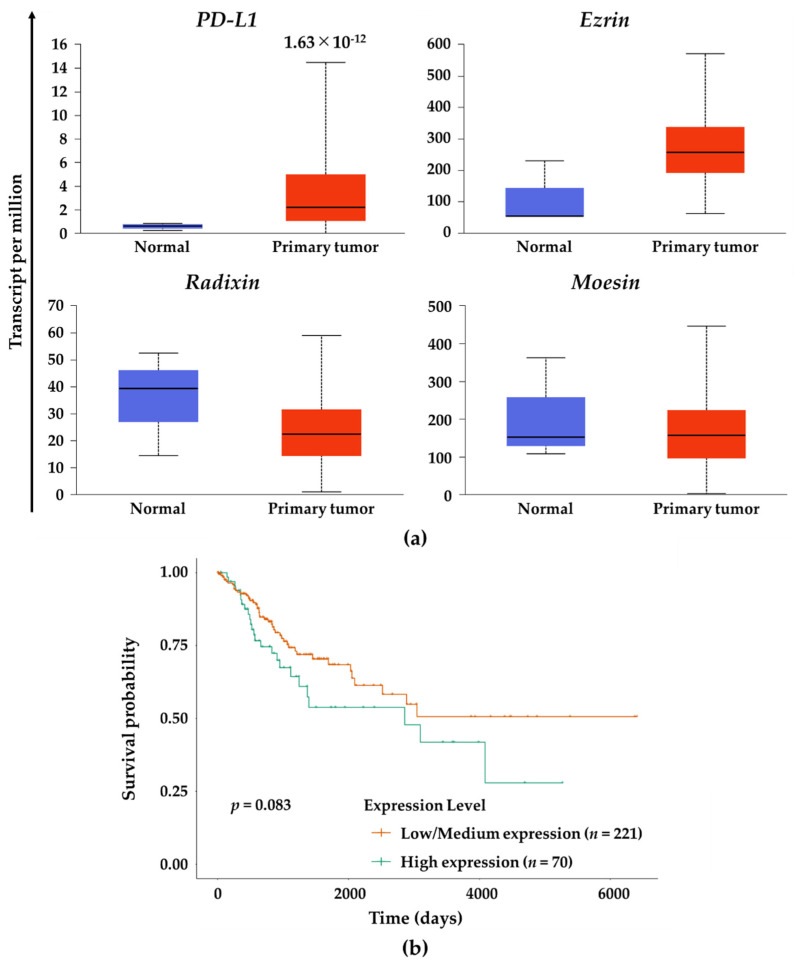
Gene expression analysis of PD-L1 and the ERM family in human cervical SCC tissue and effects of moesin expression on the survival probability of patients with cervical SCC. (**a**) Gene expression levels of PD-L1, ezrin, radixin, and moesin in the human cervical SCC tissues (*n* = 305) and normal tissues (*n* = 3) using RNA-sequencing (seq) data obtained from The Cancer Genome Atlas (TCGA) database. The estimated gene expression values based on RNA-seq data are presented as transcripts per million. All data were expressed as boxplots. (**b**) Survival probability curves for the patients with cervical SCC who had moesin expression at high (*n* = 70) or at low/medium (*n* = 221) levels.

## Data Availability

The datasets used and analyzed during this study are available from Cancer Cell Line Encyclopedia (https://sites.broadinstitute.org/ccle/datasets, accessed on 1 March 2022), DepMap, Broad (2022): DepMap 22Q1 Public. figshare. Dataset (https://doi.org/10.6084/m9.figshare.19139906.v1, accessed on 1 March 2022). Other data are contained within the article and Appendix A.

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
