# Peer review of "Moesin Serves as Scaffold Protein for PD-L1 in Human Uterine Cervical Squamous Carcinoma Cells"

_jcm, 2022, doi:10.3390/jcm11133830_

Round 1

Reviewer 1 Report

In this manuscript, Doukuni et al. tried to investigate the co-localization of ERM and PD-L1 in BOKU cells, and found that PD-LI co-localized with these three ERM proteins in the plasma membrane, and knockdown of moesin reduced the plasma membrane expression of PD-L1, suggesting that moesin maybe a scaffold protein of PD-L1 in human uterine cervical SCC. However, the authors have recently reported that ERM proteins post-translationally regulate the cell surface localization of PD-L1 by serving as scaffold proteins in different manners in several types of human cancer cells. Therefore, the innovation of this study is not outstanding enough. Additionally, there are minor defects which need further modification.

Comments:

1.       There are many kinds of SCC cell lines, the author should verify the results in serval SCC cell lines.

2.       Introduction: the author should introduce these three ERM proteins in the section of Introduction, such as cell origin, structure and function.

3.       The expression and correlation of three ERM proteins and PD-L1 in human uterine cervical tissues should be analyzed.

4.       The effect of siRNA of moesin plus PD-L1 on cell viability should be evaluated.

5.       Discussion: The possible mechanism of moesin on plasma membrane expression of PD-L1 in BOKU cells should be discussed.

6.       The mechanisms for different effects of these ERM proteins on PD-L1 in SCC should be discussed.

7.       The relationship of moesin and other regulatory factors of PD-L1 in SSC should be discussed.

Author Response

Response to Reviewer 1’s Comments

We would like to thank #Reviewer 1 for many valuable suggestions on our manuscript. We have carefully read your comments and suggestions and have made the corrections in the revised version of manuscript. Detailed responses to your comments are listed below, and we highlighted all changes with word track changes in the file labeled ‘Revised Manuscript with Track Changes’. We hope this revised manuscript would be satisfactory for publication in Journal of Clinical Medicine.

  1. There are many kinds of SCC cell lines, the author should verify the results in serval SCC cell lines.

Reply Comments.

We would like to appreciate #Reviewer 1’s valuable suggestion. According to #Reviewer 1’s comment, we have added the experimental data using other human uterine cervical SCC cell line (HCS-2 cells) we have already obtained. All the data were incorporated into the Results section and Supplementary Materials as Figure S3-S7. In addition, we have incorporated the limitation of this study into the Discussion section.

Results (Line 187–189)

Although the relative mRNA expression level of PD-L1 was low in HCS-2 cell, those of ezrin, radixin, and moesin in HCS-2 cells rank high among human uterine cervical squamous cancer cell lines analyzed (Figure 1a).

Results (Line 190–193)

Next, the relative mRNA and protein expression levels of PD-L1, ezrin, radixin, and moesin in BOKU cells and HCS-2 cells were examined using RT-PCR and western blotting, respectively. PD-L1, ezrin, radixin, and moesin mRNA and protein expression were detected in BOKU cells (Figure 1b,c) and also in HCS-2 cells (Figure S3a,b).

Results (Line 208–216)

The subcellular localization of PD-L1 and ERM proteins in BOKU cells and HCS-2 cells was scrutinized using immunofluorescence CLSM. No fluorescence signals were observed in cells incubated with goat anti-rabbit IgG secondary antibodies conjugated with Alexa Fluor 488 or Alexa Fluor 594 without primary antibodies (Figure S2a,b). The fluorescence signals of PD-L1, ezrin, radixin, and moesin in BOKU cells (Figure 2a–d) and HCS-2 cells (Figure S4a–d) were highly colocalized with that of F-actin, a representative cellular plasma membrane marker, implicating specific plasma membrane localization of these proteins in the human uterine cervical squamous cancer cells. Interestingly, double immunofluorescence staining results demonstrated that PD-L1 was highly colocalized with the ERM proteins, especially in the plasma membrane of BOKU cells (Figure 3a–c) and HCS-2 cells (Figure S5a-c).

Results (Line 235–238)

Treatment of BOKU cells and HCS-2 cells with siRNAs against ezrin, radixin, and moesin effectively significantly suppressed the mRNA levels of the respective targets (Figure 4a–c; Figure S6a), without affecting the cell viability of BOKU cells (Figure 4d) and HCS-2 cells (Figure S6b).

Results (Line 253–260)

We evaluated the effects of knockdown of ezrin, radixin, and moesin on the mRNA and cell surface expression levels of PD-L1 in BOKU cells and HCS-2 cells. Treatments of cells with the ERM siRNAs had no effects on the mRNA levels of PD-L1 as compared with NC siRNA (Figure 5a; Figure S7a). Flow-cytometric analysis showed that moesin siRNA, but not ezrin and radixin siRNAs, significantly reduced the cell surface expression of PD-L1 to the same extent as PD-L1 siRNA did in BOKU cells (Figure 5b,c) and HCS-2 cells (Figure S7b,c). Co-immunoprecipitation assays demonstrated the molecular interaction between PD-L1 and moesin in BOKU cell (Figure 5d) and HCS-2 cells (Figure S7d).

Discussion (Line 311–312)

Additionally, we also verified the similar results in HCS-2 cells.

Discussion (Line 332–335)

These results imply that moesin may be responsible for the cell surface localization of PD-L1 in BOKU cells and HCS-2 cells, in which all three ERM proteins are highly expressed.

Discussion (Line 380–386)

The limitation of this study is that in vitro relationship between PD-L1 and moesin in BOKU cells does not fully mimic the clinical patients with uterine cervical SCC received ICB therapy. Additionally, we only used two human uterine cervical SCC cell lines, although there are many kinds of human uterine cervical SCC cell lines. Therefore, we should address these issues with other human uterine cervical SCC cell lines and also with in vivo xenograft model mice to demonstrate the relationship between PD-L1 and moesin in human uterine cervical SCC.

Supplementary Materials (Line 26–68)

Figure S3. Gene and protein expression profiles of programmed death ligand-1 (PD-L1), ezrin, radixin, and moesin (ERM) in HCS-2 cells. (a) Representative amplification curves for PD-L1 and the ERM proteins as well as β-actin, included as an internal control, in HCS-2 cells as determined by real-time reverse transcription polymerase chain reaction (RT-PCR). (b)  Representative immunoblot images for PD-L1 and the ERM proteins as well as glyceraldehyde-3-phosphate dehydrogenase (GAPDH), included as an internal control, in whole-cell lysates of HCS-2 cells. Molecular weights are denoted in kDa. Data are representative of three independent experiments using at least three independent samples of total RNA and protein extracts.

Figure S4. Confocal laser scanning microscopy (CLSM) analysis of the intracellular distribution of PD-L1 and the ERM family in HCS-2 cells. (a) PD-L 1, (b) ezrin, (c) radixin, and (d) moesin labeled with Alexa Fluor 488 (green) physically colocalized with F-actin labeled with tetramethylrhodamine (red). Scale bars: 50 µm. All images are representative of at least three independent experiments.

Figure S5. Colocalization of PD-L1 with the ERM family in the plasma membrane of HCS-2 cells. CLSM analysis showed that PD-L1 labeled with Alexa Fluor 488 (green) strongly colocalized with (a) ezrin, (b) radixin, and (c) moesin labeled with Alexa Fluor 594 (red) in the plasma membrane. Scale bar 50 µm. All images are representative of at least three independent experiments.

Figure S6. Effects of RNA interference-mediated knockdown of the ERM family on target mRNA levels in and cell viability of HCS-2 cells. Cells were cultured with transfection medium (Untreated), transfection reagent (Lipofectamine), nontargeting control (NC) siRNA, or the siRNAs against the target genes at 5 nM for three days. (a) mRNA levels of ezrin, radixin, and moesin normalized to that of β-actin and relative to that of Lipofectamine alone as determined by RT-PCR. n = 3, ***p < 0.001 vs. Lipofectamine. All data are expressed as the mean ± standard of the mean (SEM) and were analyzed by one-way analysis of variance (ANOVA) followed by Dunnett’s tests. (b) Cell viability of HCS-2 cells. n = 6, ***p < 0.001 vs. Lipofectamine. All data are expressed as the mean ± SEM and were analyzed by one-way ANOVA followed by Dunnett’s tests.

Figure S7. Moesin is a scaffold protein responsible for the cell surface expression of PD-L1 in HCS-2 cells. Cells were cultured with transfection medium (Untreated), transfection reagent (Lipofectamine), NC siRNA, or the siRNAs against the target genes at 5 nM for three days (a–c). (a) PD-L1 mRNA expression levels normalized to that of β-actin in cells relative to that in Lipofectamine-treated cells as determined by RT-PCR; n = 3, * p < 0.05 vs. Lipofectamine. (b) Mean fluorescence intensity of allophycocyanin (APC)-labeled PD-L1 in the surface plasma membrane relative to that in Lipofectamine treatment, quantified by flow cytometry; n = 3, *** p < 0.001, * p < 0.05 vs. Lipofectamine. All data are expressed the mean ± SEM and were analyzed by one-way ANOVA followed by Dunnett’s tests. (c) Overlay of representative histograms of the mean fluorescence intensity of APC-PD-L1 in the surface plasma membrane of HCS-2 cells as measured by flow cytometry. (d) Representative immunoblots of moesin and PD-L1 in whole-cell lysates (input) and immune precipitates pulled down with a control IgG or an anti-PD-L1 Ab. Molecular weight is expressed in kDa.

  1. Introduction: the author should introduce these three ERM proteins in the section of Introduction, such as cell origin, structure and function.

Reply Comments.

We would like to appreciate #Reviewer 1’s valuable suggestion. According to #Reviewer 1’s comment, we have incorporated the sentence describing the cell origin, structure and function of three ERM proteins into the Introduction.

Introduction (Line 55–67)

Ezrin/radixin/moesin (ERM) protein family members are three closely related proteins distributed as scaffold proteins in a whole body [17]. Ezrin was discovered in the chicken intestinal brush borders as a component of microvilli [18]. Radixin was purified from the undercoat of the adherens junction isolated from the rat liver [19]. Moesin was isolated from bovine uteri enriched in smooth muscle cells as a heparin-binding protein [20]. ERM proteins consist of a 4.1-band ERM (FERM) domain at the amino-terminus, a central α-helical domain located between the amino-terminus and carboxy-terminus domains, and a F-actin-binding domain at the carboxy-terminus [21]. The FERM domain normally interacts with the intramolecular carboxy-terminus in the cytosol, leading to dormant inactive conformation of ERM proteins that is incapable of interacting with other proteins, while the intramolecular binding is dissociated by phosphorylation of threonine on the carboxy-terminus [21-23], enable ERM proteins to be active conformation that can play multiple roles in cell motility, cell adhesion, and signal transduction [17,24-26].

References (Line 461–482)

  1. Sato, N.; Funayama, N.; Nagafuchi, A.; Yonemura, S.; Tsukita, S.; Tsukita, S. A gene family consisting of ezrin, radixin and moesin. Its specific localization at actin filament/plasma membrane association sites. J. Cell Sci. 1992, 103 ( Pt 1), 131-143. https://doi.org/10.1242/jcs.103.1.131.
  2. Bretscher, A. Purification of an 80,000-dalton protein that is a component of the isolated microvillus cytoskeleton, and its localization in nonmuscle cells. J. Cell Biol. 1983, 97, 425-432. https://doi.org/10.1083/jcb.97.2.425.
  3. Tsukita, S.; Hieda, Y. A new 82-kD barbed end-capping protein (radixin) localized in the cell-to-cell adherens junction: purification and characterization. J. Cell Biol. 1989, 108, 2369-2382.
  4. Lankes, W.; Griesmacher, A.; Grunwald, J.; Schwartz-Albiez, R.; Keller, R. A heparin-binding protein involved in inhibition of smooth-muscle cell proliferation. Biochem. J. 1988, 251, 831-842.
  5. Tsukita, S.; Yonemura, S. Cortical actin organization: lessons from ERM (ezrin/radixin/moesin) proteins. J. Biol. Chem. 1999, 274, 34507-34510.
  6. Tsukita, S.; Oishi, K.; Sato, N.; Sagara, J.; Kawai, A. ERM family members as molecular linkers between the cell surface glycoprotein CD44 and actin-based cytoskeletons. J. Cell Biol. 1994, 126, 391-401.
  7. Matsui, T.; Maeda, M.; Doi, Y.; Yonemura, S.; Amano, M.; Kaibuchi, K.; Tsukita, S. Rho-kinase phosphorylates COOH-terminal threonines of ezrin/radixin/moesin (ERM) proteins and regulates their head-to-tail association. J. Cell Biol. 1998, 140, 647-657.
  8. Fehon, R.G.; McClatchey, A.I.; Bretscher, A. Organizing the cell cortex: the role of ERM proteins. Nat. Rev. Mol. Cell Biol. 2010, 11, 276-287. https://doi.org/10.1038/nrm2866.
  9. Bretscher, A.; Edwards, K.; Fehon, R.G. ERM proteins and merlin: integrators at the cell cortex. Nat. Rev. Mol. Cell Biol. 2002, 3, 586-599. https://doi.org/10.1038/nrm882.
  10. Bretscher, A.; Chambers, D.; Nguyen, R.; Reczek, D. ERM-Merlin and EBP50 protein families in plasma membrane organization and function. Annu. Rev. Cell Dev. Biol. 2000, 16, 113-143. https://doi.org/10.1146/annurev.cellbio.16.1.113.

  1. The expression and correlation of three ERM proteins and PD-L1 in human uterine cervical tissues should be analyzed.

Reply Comments.

We would like to appreciate #Reviewer 1’s valuable suggestion. According to #Reviewer 1’s comment, we have incorporated the expression and correlation of three ERM proteins and PD-L1 in human uterine cervical squamous cell carcinoma tissues analyzed by using The Cancer Genome Atlas (TCGA) datasets into the Materials and Methods, Results, and Discussion.

Materials and Methods (Line 166–172)

2.9. Analysis for Gene Expression of PD-L1 and the ERM Family in Patients with Cervical SCC

Gene expression levels PD-L1 and the ERM family in the tumor tissues derived from patients with cervical SCC and impacts of moesin expression level on the patient survival probabilities were assessed with UALCAN. UALCAN is a web portal enables evaluating gene expression and survival analysis on approximately 20,500 protein-coding genes in 33 different tumor types using RNA-sequencing data obtained from The Cancer Genome Atlas (TCGA) project [39, 40].

Results (Line 289–306)

3.6. Gene Expression Analysis of PD-L1 and the ERM Family in Human Cervical SCC Tissue

Finally, we assessed gene expression levels PD-L1 and the ERM family in the tu-mor tissues derived from patients with cervical SCC. Gene expression of PD-L1 was significantly higher in the human cervical SCC tissues relative to normal tissues, while there were no significant changes in the gene expression of the ERM family (Figure 7a). In contrast, no correlations were observed between PD-L1 and three ERM gene expressions. Importantly, survival probability in the cervical SCC patients with higher expression of moesin was not significantly (p = 0.083) but obviously lower than those with low/medium expression of moesin (Figure 7b).

Figure 7. Gene expression analysis of PD-L1 and the ERM family in human cervical SCC tissue and effects of moesin expression on the survival probability of patients with cervical SCC. (a) Gene expression levels of PD-L1, ezrin, radixin, and moesin in the human cervical SCC tissues (n = 305) and normal tissues (n = 3) using RNA-sequencing (seq) data obtained from The Cancer Genome Atlas (TCGA) database. The estimated gene expression values based on RNA-seq data are presented as transcripts per million. All data were expressed as boxplots. (b) Survival probability curves for the patients with cervical SCC who had moesin expression at high (n = 70) or at low/medium (n = 221) levels.

Discussion (Line 359–362)

In fact, analysis using the human cervical SCC tissues obtained from TCGA database showed a higher expression of PD-L1 gene in the human cervical SCC tissue and a lower survival rate in the cervical SCC patients who had a higher expression of moesin.

References (Line 513–518)

  1. Chandrashekar, D.S.; Karthikeyan, S.K.; Korla, P.K.; Patel, H.; Shovon, A.R.; Athar, M.; Netto, G.J.; Qin, Z.S.; Kumar, S.; Manne, U.; et al. UALCAN: An update to the integrated cancer data analysis platform. Neoplasia 2022, 25, 18-27. https://doi.org/10.1016/j.neo.2022.01.001.
  2. Chandrashekar, D.S.; Bashel, B.; Balasubramanya, S.A.H.; Creighton, C.J.; Ponce-Rodriguez, I.; Chakravarthi, B.; Varambally, S. UALCAN: A Portal for Facilitating Tumor Subgroup Gene Expression and Survival Analyses. Neoplasia 2017, 19, 649-658. https://doi.org/10.1016/j.neo.2017.05.002.

  1. The effect of siRNA of moesin plus PD-L1 on cell viability should be evaluated.

Reply Comments.

We would like to appreciate #Reviewer 1’s valuable suggestion. Unfortunately, the concentrations (5 nM) of each siRNA used in this study is maximus dose without exhibiting the cytotoxicity with the knockdown activity for each target gene in our experimental models. If cells were exposed to moesin siRNA 5 nM combined with PD-L1 siRNA 5 nM (total of 10 nM siRNAs) even to 10 nM of non-targeting control siRNA, cell viability seems to be decreased in comparison with the control. Therefore, we have employed the 5 nM of siRNAs for knockdown of each target gene in Figure 4-5.

  1. Discussion: The possible mechanism of moesin on plasma membrane expression of PD-L1 in BOKU cells should be discussed.

Reply Comments.

We would like to appreciate #Reviewer 1’s valuable suggestion. According to #Reviewer 1’s comment, we have incorporated the possible mechanism of moesin to regulate the plasma membrane expression of PD-L1 in BOKU cells into the Discussion.

Discussion (Line 337–343)

Recently, Meng et al. reported that in the human breast cancer cell line p-moesin pre-vents PD-L1 from the ubiquitination that leads to the proteasomal degradation by competing with E3 ubiquitin ligase, thereby inhibiting degradation of PD-L1, which in turn, stabilizes PD-L1 in the surface plasma membrane [64]. These observation raises the possibility that p-moesin may contribute to the plasma membrane stabilization of PD-L1 in BOKU cells, possibly by preventing the ubiquitin-mediated proteasomal degradation.

  1. The mechanisms for different effects of these ERM proteins on PD-L1 in SCC should be discussed.

Reply Comments.

We would like to appreciate #Reviewer 1’s valuable suggestion. According to #Reviewer 1’s comment, we have incorporated the possible mechanisms for different effects of these ERM proteins on PD-L1 in SCC into the Discussion.

Discussion (Line 343–356)

On the other hand, the mechanism by which ERM proteins differently influences the plasma membrane localization of PD-L1 in several cancer cell types remains unknown. One possibility is that among ERM proteins, the principal partner for PD-L1 may be attributed, at least in part, to a large variation of ERM expression profile according to the histological cancer types and/or cancer sites. A growing body of evidence suggests that radixin predominantly regulates the plasma membrane localization of multidrug resistance protein 2, an efflux transporter abundant in hepatocyte [65-70]. This is likely because radixin is a dominant ERM protein in the hepatic tissues and hepatocytes [71]. Given that in contrast to ezrin and radixin, moesin is highly expressed in the human uterine cervix SCC [58-60], moesin may regulate the plasma membrane localization of PD-L1 as a predominant ERM protein in BOKU cells and HCS-2 cells, as is the case with ezrin in HeLa and LS180 cells [31,32]. The detailed mechanism by which PD-L1 selects a particular ERM protein as a partner for its plasma membrane localization in several cancer cell types will have to be addressed in future studies.

  1. The relationship of moesin and other regulatory factors of PD-L1 in SSC should be discussed.

Reply Comments.

We would like to appreciate #Reviewer 1’s valuable suggestion. According to #Reviewer 1’s comment, we have incorporated the relationship of moesin and other regulatory factors of PD-L1 in SSC, however, that is common with the possible mechanism of moesin to regulate the plasma membrane expression of PD-L1 into the Discussion.

Discussion (Line 337–343)

Recently, Meng et al. reported that in the human breast cancer cell line p-moesin pre-vents PD-L1 from the ubiquitination that leads to the proteasomal degradation by competing with E3 ubiquitin ligase, thereby inhibiting degradation of PD-L1, which in turn, stabilizes PD-L1 in the surface plasma membrane [64]. These observation raises the possibility that p-moesin may contribute to the plasma membrane stabilization of PD-L1 in BOKU cells, possibly by preventing the ubiquitin-mediated proteasomal degradation.

Reviewer 2 Report

The authors studied moesin and its scaffold protein complex for PD-L1 in human uterine cervical squamous carcinoma cells. However, this study is somewhat similar to their previous studies already published in other journals in MDPI. The scientific novelty of this work is very low or none.

The quality of the figure presentation is poor with low maginication. I can't appreciate the author's claim in Figures 2 and 3. 

In Figure 5a, the authors presented treatment of the ERM siRNAs had no effects on the mRNA levels of PD-L1 as compared to NC siRNA, however, figure 5a showed siRNA-ERM showed a decrease in PD-L1 mRNA. In Figure 5b, siRNA-ERM doesn't show any difference in cell count, but I wonder why the authors used flow cytometry to detect cell counts instead of using a traditional way of in vitro counting cells after siRNA treatment.

Is there any binding activity between Radixin and Ezrin with PD-L1?

What is the clinical implication of this study?

The authors should study whether the loss of interaction between Moesin and PD-L1 will impact the immune response of uterine cervical cancer in vivo murine models.

Author Response

Response to Reviewer 2’s Comments

We would like to thank #Reviewer 2 for the greatest evaluation on our manuscript. We have carefully read your comments and suggestions and have made the corrections in the revised version of manuscript. Detailed responses to your comments are listed below, and we highlighted all changes with word track changes in the file labeled ‘Revised Manuscript with Track Changes’. We hope this revised manuscript would be satisfactory for publication in Journal of Clinical Medicine.

  1. The quality of the figure presentation is poor with low magnification. I can't appreciate the author's claim in Figures 2 and 3.

Reply Comments.

We sincerely apologize for the inconvenience we caused #Reviewer 2 with the quality of Figure 2 and 3. According to #Reviewer 2’s comment, we have added enlarged image of each Figure for merged images in the Figure 2 and 3.

Results (Line 221–222)

Legend for Figure 2

Higher magnification images in the rightmost are from the corresponding white rectangle region in the merged panels. Scale bars: 20 µm.

Results (Line 228–229)

Legend for Figure 3

Higher magnification images in the rightmost are from the corresponding white rectangle region in the merged panels. Scale bars: 20 µm.

  1. In Figure 5a, the authors presented treatment of the ERM siRNAs had no effects on the mRNA levels of PD-L1 as compared to NC siRNA, however, figure 5a showed siRNA-ERM showed a decrease in PD-L1 mRNA. In Figure 5b, siRNA-ERM doesn't show any difference in cell count, but I wonder why the authors used flow cytometry to detect cell counts instead of using a traditional way of in vitro counting cells after siRNA treatment.

Reply Comments.

We sincerely apologize for the inconvenience we caused #Reviewer 2 with the interpretation of Figure 5a-c. In Figure 5b-c, we have performed flowcytometric analysis to measure the cell surface expression levels of PD-L1 but not to detect the cell counts like cell viability assay. We speculate that the position of (c) in Figure 5 was quite far away from corresponding column that seems to give rise to the confusion. Our apologies for this confusion. According to #Reviewer 2’s comment, we have made the corrections in Figure 5 to understand clearly for every reader.

Results (Line 264–265)

Figure 5

  1. Is there any binding activity between Radixin and Ezrin with PD-L1?

Reply Comments.

We would like to appreciate #Reviewer 2’s valuable suggestion. According to #Reviewer 2’s comment, we have conducted additional experiments to detect the protein-protein interaction between PD-L1 and Ezrin and Radixin.

Results (Line 259–262)

Co-immunoprecipitation assays demonstrated the molecular interaction between PD-L1 and moesin in BOKU cell (Figure 5d) and HCS-2 cells (Figure S7d) in addition to the molecular interaction between PD-L1 and ezrin, radixin in BOKU cells (Figure S8).

Supplementary Materials (Line 70–74)

Figure S3. Molecular interaction of PD-L1 with ezrin and radixin in BOKU cells. Representative immunoblots of ezrin and radixin in whole-cell lysates (input) and immune precipitates pulled down by a control IgG or an anti-PD-L1 antibody. Molecular weight is expressed in kDa.

  1. What is the clinical implication of this study?

Reply Comments.

We would like to appreciate #Reviewer 2’s valuable suggestion. According to #Reviewer 2’s comment, we have incorporated the sentence describing our possible therapeutic strategy to mention the clinical implication of this study in the Discussion.

Discussion (Line 364–370)

Our therapeutic strategy that siRNA-mediated inhibition of moesin decreases the cell surface localization of PD-L1 from intracellular compartments seems to be an attractive therapeutic modality independent of the mutation in the extracellular region of PD-L1 protein where the target sequence of Abs against PD-L1 is present, which is the cause of tolerance to the PD-L1 Abs. Therefore, one possible advantage of this method is that it can be applicable to the patients developed tolerance to the existing Abs against PD-L1.

  1. The authors should study whether the loss of interaction between Moesin and PD-L1 will impact the immune response of uterine cervical cancer in vivo murine models.

Reply Comments.

We would like to appreciate #Reviewer 2’s valuable suggestion. As #Reviewer 2 pointed out, we also think it is very important to ascertain whether the results of our present in vitro experiments are also replicated in the in vivo murine models. In the future study, we should address this issue using in vivo xenograft mice model to investigate whether loss of interaction between PD-L1 and moesin influence the anticancer immune response. According to #Reviewer 2’s fruitful advice, we have incorporated our future plan and the limitation of this study in the Discussion section.

Discussion (Line 371–373)

In order to ascertain whether the results of our present in vitro data are also observed in the in vivo experiments employed xenograft model mice should be performed in our future studies.

Discussion (Line 380–386)

The limitation of this study is that in vitro relationship between PD-L1 and moesin in BOKU cells does not fully mimic the clinical patients with uterine cervical SCC received ICB therapy. Additionally, we only used two human uterine cervical SCC cell lines, although there are many kinds of human uterine cervical SCC cell lines. Therefore, we should address these issues with other human uterine cervical SCC cell lines and also with in vivo xenograft model mice to demonstrate the relationship between PD-L1 and moesin in human uterine cervical SCC.

Reviewer 3 Report

Needs tightening. What are limitations, if any, to other methods. What is the advantage of this technology?

Author Response

Response to Reviewer 3’s Comments

We would like to thank #Reviewer 3 for the greatest evaluation on our manuscript. We have carefully read your comments and suggestions and have made the corrections in the revised version of manuscript. Detailed responses to your comments are listed below, and we highlighted all changes with word track changes in the file labeled ‘Revised Manuscript with Track Changes’. We hope this revised manuscript would be satisfactory for publication in Journal of Clinical Medicine.

  1. What are limitations, if any, to other methods.

Reply Comments.

We would like to appreciate #Reviewer 3’s valuable suggestion. According to #Reviewer 3’s comment, we have incorporated the limitation of this study in the Discussion.

Discussion (Line 371–373)

In order to ascertain whether the results of our present in vitro data are also observed in the in vivo experiments employed xenograft model mice should be performed in our future studies.

Discussion (Line 380–386)

The limitation of this study is that in vitro relationship between PD-L1 and moesin in BOKU cells does not fully mimic the clinical patients with uterine cervical SCC received ICB therapy. Additionally, we only used two human uterine cervical SCC cell lines, although there are many kinds of human uterine cervical SCC cell lines. Therefore, we should address these issues with other human uterine cervical SCC cell lines and also with in vivo xenograft model mice to demonstrate the relationship between PD-L1 and moesin in human uterine cervical SCC.

  1. What is the advantage of this technology?

Reply Comments.

We would like to appreciate #Reviewer 3’s valuable suggestion. According to #Reviewer 3’s comment, we have incorporated the advantage of this technology in the Discussion.

Discussion (Line 362–367)

Our therapeutic strategy that siRNA-mediated inhibition of moesin decreases the cell surface localization of PD-L1 from intracellular compartments seems to be an attractive therapeutic modality independent of the mutation in the extracellular region of PD-L1 protein where the target sequence of Abs against PD-L1 is present, which is the cause of tolerance to the PD-L1 Abs. Therefore, one possible advantage of this method is that it can be applicable to the patients developed tolerance to the existing Abs against PD-L1.

Round 2

Reviewer 1 Report

The authors have addressed all concerns. 

Reviewer 2 Report

No further comments.